# Implications of W-Boson Mass Anomaly for Atomic Parity Violation

Hoang Bao Tran Tan * and Andrei Derevianko

Department of Physics, University of Nevada, Reno, NV 89557, USA
* Correspondence: htrantan@unr.edu

**Abstract:** We consider the implications of the recent measurement of the W-boson mass $M_W = 80{,}433.5 \pm 9.4\,\mathrm{MeV/c^2}$ for atomic parity violation experiments. We show that the change in $M_W$ shifts the Standard Model prediction for the $^{133}$Cs nuclear weak charge to $Q_W(^{133}\mathrm{Cs}) = -73.11(1)$, i.e., by $8.5\sigma$ from its current value, and the proton weak charge by 2.7%. The shift in $Q_W(^{133}\mathrm{Cs})$ ameliorates the tension between existing determinations of its value and motivates more accurate atomic theory calculations, while the shift in $Q_W(p)$ inspires next-generation atomic parity violation experiments with hydrogen. Using our revised value for $Q_W(^{133}\mathrm{Cs})$, we also readjust constraints on parameters of physics beyond the Standard Model. Finally, we reexamine the running of the electroweak coupling for the new W boson mass.

**Keywords:** W-boson mass anomaly; atomic parity violation; physics beyond the standard model

## 1. Introduction

Atomic parity violation (APV) is a major means for testing the electroweak (EW) sector of the Standard Model (SM) at low energy. Currently, the best APV result provides a confirmation of the SM prediction of the $^{133}$Cs nuclear weak charge at the level of 0.35% accuracy [1]. Future APV experiments with expected accuracy 0.1–0.2% [2–9] may help resolve the tension between high-energy Z-pole measurements of $\sin^2 \theta_W$ (here $\theta_W$ is the weak mixing angle) [10,11] when extrapolated to the APV scale. APV experiments are also uniquely sensitive to a certain class of new physics to which high-energy probes are blind.

The use of APV to constrain physics beyond the SM relies on precise measurement of the APV amplitude $E_{\mathrm{PV}}$, accurate theoretical calculation of the atomic structure factor $k_{\mathrm{PV}}$ needed for extracting the nuclear weak charge $Q_W$ via $E_{\mathrm{PV}} = k_{\mathrm{PV}}Q_W$, and exact knowledge of the SM prediction for $Q_W$ against which the experimentally extracted value is compared. The most accurate measurement of $E_{\mathrm{PV}}$ comes from the Boulder group for $^{133}$Cs with an uncertainty of 0.35% [1], although a new experiment is being planned with an aim of a 0.2% accuracy [4,5].

Early atomic calculations of $k_{\mathrm{PV}}$ for $^{133}$Cs at the level of 0.4% uncertainty [12–15] gave a value of $Q_W$ that is $2.5\sigma$ away from the SM prediction. Later developments resulted in the inclusion of sub-1% contributions from Breit and QED corrections and culminated in the most detailed coupled-cluster singles doubles and valence triples calculation (CCSDvT) with an uncertainty of 0.27% and a value for $Q_W$ in an essential agreement with the SM [16]. A more recent reevaluation yielded a $Q_W$, which is $1.5\sigma$ away from the SM value whilst raising the theoretical uncertainty back to 0.5% [17]. The latest 0.3%-accurate calculation [18–20] gives a result agreeing with Ref. [16]. A new parity-mixed coupled-cluster approach to calculating $k_{\mathrm{PV}}$ is under development [21], with a goal of reducing the theoretical uncertainty to 0.2%.

Once the values of $E_{\mathrm{PV}}$ and $k_{\mathrm{PV}}$ are known, the nuclear weak charge $Q_W$ may be extracted using $E_{\mathrm{PV}} = k_{\mathrm{PV}}Q_W$ and compared with the SM prediction. A disagreement between the two results could provide hints about physics beyond the SM. Within the SM

itself, $Q_W$ is expressed in terms of the axial–electron–vector–nucleon coupling constants $g_{AV}^{ep}$ and $g_{AV}^{en}$. At the tree level, these coupling constants depend on the weak mixing angle, with one-loop and leading two-loop corrections coming from the $W$ and $Z$ boson self-energies, the $\gamma Z$ mixing renormalization of $\sin^2 \theta_W$, and the so-called $WW$, $ZZ$, and $\gamma Z$ box diagrams [22–25]. The low-energy value of $\sin^2 \theta_W$ may be obtained from the measured $Z$-pole value by using its scale dependence. New physics contributions to the weak charge $Q_W$ may arise from multiple mechanisms: (i) tree-level exchange of a new $Z'$ boson with mass at the TeV scale [26,27], (ii) corrections to the $W$ and $Z$ boson self-energies due to vacuum polarization involving beyond-SM particles [26,28], (iii) kinetic and mass mixing of a "dark" $Z_d$ boson of mass $\sim$ MeV $-$ GeV with the photon and the $Z$ boson [29–31], or (iv) an oscillating $\bar{\theta}_{\text{QCD}}$ term in the form of interaction with the axion and axion-like particles [32].

The reference SM value for the weak mixing angle is usually obtained from global fits of electroweak observables such as masses and widths of the $Z$ and $W$ bosons as well as left-right and forward-backward asymmetries in a variety of scattering processes involving the weak interaction. In this paper, we focus on the implications of the recently reported $W$ boson mass from the CDF II collaboration, which shows a $7\sigma$ deviation from the current global fit value [33]. We show that the CDF II stand-alone result implies a shift in the $Z$-pole value of the weak mixing angle and thereby modifies the SM value of the $^{133}$Cs nuclear weak charge $Q_W$. Similarly, it shifts the value of the proton weak charge by 2.7%, further motivating APV experiments in hydrogen. By using the new value of $Q_W^{\text{SM}}(^{133}\text{Cs})$ implied by the $W$ boson mass anomaly [33] and existing APV results for $^{133}$Cs, we readjust limits on the ratio of the coupling-to-mass of the new $Z'$ boson, the weak isospin-conserving parameter of vacuum polarization effects on gauge boson propagators, and on the parameters describing the SM couplings to a dark $Z_d$ boson. Implications of the new $W$ boson mass measurement [33] for other physics beyond the SM scenarios were considered in Refs. [34–43].

## 2. Theory

### 2.1. Electroweak Phenomenology and Atomic Parity Violation

The electroweak (EW) sector of the standard model is described in terms of the $SU(2)_L \times U(1)_Y$ gauge group with corresponding vector fields $W_\mu^i$ ($i = 1, 2, 3$) and $B_\mu$ with gauge couplings $g$ and $g'$ (see, e.g., Refs. [44,45]). Spontaneous breaking of the EW gauge symmetry is effected by introducing a complex scalar Higgs doublet $\phi$ with a Lagrangian

$$\mathcal{L}_\phi = (D_\mu \phi)^\dagger (D^\mu \phi) + \mu^2 \phi^\dagger \phi + \frac{\lambda^2}{2} (\phi^\dagger \phi)^2 \,, \tag{1}$$

where the covariant derivative is defined as

$$D_\mu \phi \equiv \left( \partial_\mu + \frac{ig}{2} \sigma_i W_\mu^i + \frac{ig'}{2} B_\mu \right) \phi \,. \tag{2}$$

Here, $\sigma_i$ are the Pauli matrices.

For $\mu^2 < 0$, the potential (1) has a minimum at $v = \sqrt{2}|\mu|/\lambda$, around which point $\phi$ may be transformed into a single real scalar field $H$ with vanishing vacuum expectation value. After such a transformation, one finds that the Lagrangian (1) contains the following terms

$$\mathcal{L}_\phi \supset \frac{1}{2} M_H^2 H^2 + M_W^2 W^{\mu -} W_\mu^+ + \frac{1}{2} M_Z^2 Z^\mu Z_\mu \,, \tag{3}$$

where

$$W_\mu^\pm \equiv \frac{W_\mu^1 \pm i W_\mu^2}{\sqrt{2}}\,, \tag{4}$$

$$Z_\mu \equiv \frac{g W_\mu^3 - g' B_\mu}{\sqrt{g^2 + g'^2}}\,, \tag{5}$$

are the charged $W$ boson and neutral $Z$ boson fields, and

$$M_H = \lambda v\,, \tag{6}$$

$$M_W = g v / 2\,, \tag{7}$$

$$M_Z = \sqrt{g^2 + g'^2}\, v / 2\,, \tag{8}$$

are the masses of the Higgs boson, $W$ boson, and $Z$ boson, respectively.

The Higgs field breaks the $SU(2)_L \times U(1)_Y$ symmetry down to an $SU(2)_{\text{weak}}$ symmetry of weak interactions mediated by the $W^\pm$ and $Z$ bosons (see Equations (4)) and a $U(1)_{\text{elec}}$ symmetry with electromagnetic interactions mediated by the photon field $A_\mu \equiv \frac{g W_\mu^3 - g' B_\mu}{\sqrt{g^2 + g'^2}}$. With this, the Lagrangian for the fermion fields $\psi_i$ reads

$$\mathcal{L}_F = \sum_i \bar{\psi}_i \left[ i \slashed{\partial} - m_i \left(1 + \frac{H}{v}\right) \right] \psi_i - \frac{g}{\sqrt{2}} \left( J_W^{\mu\dagger} W_\mu^+ + J_W^\mu W_\mu^- + J_A^\mu A_\mu + J_Z^\mu Z_\mu \right), \tag{9}$$

where $m_i$ is the fermion mass and $\slashed{\partial} \equiv \gamma^\mu \partial_\mu$. The definitions for the weak charged current $J_W^\mu$, the weak neutral current $J_Z^\mu$, and the electromagnetic current $J_A^\mu$ may be found, e.g., in Ref. [46]. For small momentum transfer $Q^2 \ll M_{W,Z}^2$, the interaction terms in Equation (9) reduce to the effective charged ($\mathcal{L}_{\text{CC}}$) and neutral current ($\mathcal{L}_{\text{NC}}$) interactions

$$\mathcal{L}_{\text{CC}} = -2 J_W^{\mu\dagger} J_{W\mu} / v^2\,, \tag{10}$$

$$\mathcal{L}_{\text{NC}} = -\cos^2\theta_W J_Z^\mu J_{Z\mu} / v^2\,, \tag{11}$$

where $G_F \equiv 1/(\sqrt{2} v^2) = g^2/(2\sqrt{2} M_W^2)$ is the Fermi constant and $\theta_W = \tan^{-1}(g'/g)$ is the Weinberg angle. The effective four-fermion interaction (11) contains a parity-violating (PV) interaction

$$\mathcal{L}_{\text{NC}}^{eq} = \frac{G_F}{\sqrt{2}} \bar{e} \gamma_\mu \gamma_5 e \left( g_{AV}^{eu} \bar{u} \gamma^\mu u + g_{AV}^{ed} \bar{d} \gamma^\mu d \right), \tag{12}$$

describing the couplings between electrons and quarks by exchange of a $Z$ boson. Here, $g_{AV}^{eu}$ and $g_{AV}^{ed}$ are the axial–electron–vector–quark coupling constants, related to the axial–electron–vector–nucleon coupling constants via $g_{AV}^{ep} = 2 g_{AV}^{eu} + g_{AV}^{ed}$ and $g_{AV}^{en} = g_{AV}^{eu} + 2 g_{AV}^{ed}$. It is this interaction that ultimately gives rise to the spin-independent APV observables. For a more comprehensive review of low-energy EW experiments, see, e.g., Ref. [47].

The EW Lagrangians (1) and (9) depend on the set of parameters $\{g, g', \mu^2, \lambda^2, m_i\}$, whose values cannot be derived algebraically from within the SM and can only be determined experimentally. For this purpose, it may be more convenient to measure other sets of derived quantities, such as $\{g, \theta_W, M_H, v, m_i\}$ or $\{M_Z, \alpha, M_W, G_F, m_i\}$, where $\alpha = e^2/(\hbar c)$ is the fine-structure constant. Among these derived parameters, the quantities $M_Z$, $G_F$, and $\alpha$ have the lowest experimental errors. Namely, $M_Z = 91.1876(21)$ GeV was determined from the $Z$ line-shape scan [48], $G_F = 1.1663787(6) \times 10^{-5}\,\text{GeV}^{-2}$ was derived from muon lifetime [49], and $\alpha = 1/137.035999084(21)$ was obtained by combining measurements of the $e^\pm$ anomalous magnetic moment [50] with measurements of the Rydberg constant and atomic masses with interferometry of atomic recoil kinematics [51,52]. As a result, we keep these fixed in our analysis below.

The quantities $\theta_W$, $M_W$, $M_H$, and $m_i$ are generally less well constrained (except for $m_{e,\mu,\tau}$). The Weinberg angle $\theta_W$, or more precisely, $\sin^2 \theta_W$, is measured in a variety of schemes, depending on the energy scale, including low-energy APV [1,53–62], PV neutrino scattering [63–66], as well as various types of asymmetries in scattering and decay processes at low energy [67–75] and high energy [10,48,76–81]. The mass $M_W$ is obtained in W-pair production or single-W production at energy $Q \sim M_Z$ [76–78]. Combining $\sin^2 \theta_W$ and $M_W$ allows one to constrain $M_H$ and the top quark mass $m_t$ via [82].

$$M_W^2 \sin^2 \theta_W = \frac{A^2}{1 - \Delta r} ,\tag{13}$$

where $A \equiv \sqrt{\pi \alpha / (\sqrt{2} G_F)}$ and $\Delta r$ includes loop corrections to $M_W$, which depend on $m_t$ and $M_H$. Alternatively, one may use direct experimental values for $m_t$ and $M_H$ to constrain $M_W$ and $\sin^2 \theta_W$.

We note that there exist in the literature several different definitions for $\sin^2 \theta_W$. At the tree level, one has

$$\sin^2 \theta_W = 1 - \frac{M_W^2}{M_Z^2} = \frac{g'^2}{g^2 + g'^2} .\tag{14}$$

One may promote the first equality in Equation (14) to a definition of the renormalized $\sin^2 \theta_W$ to all orders in perturbation theory (the so-called on-shell scheme). In this case, the radiative correction $\Delta r$ has a quadratic dependence on $m_t$,

$$\Delta r \approx 1 - \frac{\alpha}{\hat{\alpha}_Z} - \frac{3 G_F m_t^2}{8\sqrt{2}\pi^2} \frac{\cos^2 \theta_W}{\sin^2 \theta_W} + \frac{11\alpha}{48\pi \sin^2 \theta_W} \ln \frac{M_H^2}{M_Z^2} ,\tag{15}$$

where $\hat{\alpha}_Z \equiv \alpha(M_Z)$ is the value of the fine structure constant at $M_Z$ and may receive large spurious contributions from higher orders $O(\alpha m_t^2 / M_W^2)$. A more popular approach promotes the second equality in Equation (14) to an $\overline{\text{MS}}$ (modified minimal subtraction) prescription with the quantity

$$\sin^2 \hat{\theta}_W(\mu) \equiv \frac{\hat{g}'^2(\mu)}{\hat{g}^2(\mu) + \hat{g}'^2(\mu)} ,\tag{16}$$

where $\hat{g}'$ and $\hat{g}$ are $\overline{\text{MS}}$ quantities, and $\mu$ is an energy scale conventionally chosen to be $M_Z$. With this $\overline{\text{MS}}$ definition, the identity (13) becomes

$$M_W^2 \sin^2 \hat{\theta}_W = \frac{A^2}{1 - \Delta \hat{r}_W} ,\tag{17}$$

where the radiative correction $\Delta \hat{r}_W$ now has no quadratic dependence on $m_t$,

$$\Delta \hat{r}_W \approx 1 - \frac{\alpha}{\hat{\alpha}_Z} + \frac{4\alpha}{48\pi \hat{s}^Z} \ln \frac{M_H^2}{M_Z^2} ,\tag{18}$$

where $\hat{s}_Z^2 \equiv \sin^2 \hat{\theta}_W(M_Z)$. The $\overline{\text{MS}}$ and on-shell definitions are related via

$$\hat{s}_Z^2 = c(m_t, M_H) \sin^2 \theta_W ,\tag{19}$$

with a multiplicative coefficient $c(m_t, M_H) = 1.0351(3)$ [82]. For APV, the quantity

$$\hat{s}_0^2 \equiv \sin^2 \hat{\theta}_W(0)$$

is relevant, where the energy scale is set to zero.

Let us now consider the EW phenomenology at low energies. In particular, we concentrate on APV parameterized by a nuclear weak charge $Q_W$. The nuclear weak charge

$Q_W$ arises as a parameter of the effective APV Hamiltonian density obtained by integrating out the quark fields in the Lagrangian density (12), obtaining

$$\mathcal{L}_{\mathrm{NC}}^{eq} \to H_W = -\frac{G_F}{2\sqrt{2}}\bar{e}(\mathbf{r})\gamma_5 e(\mathbf{r})Q_W\rho(\mathbf{r}),\tag{20}$$

where $\rho(r)$ is the nuclear density. The nuclear weak charge $Q_W$ receives coherent contributions from both protons and neutrons and may be written, at the tree level, as [82].

$$Q_W = Z\left(1 - 4\hat{s}_0^2\right) - N,\tag{21}$$

where $Z$ is the atomic number, and $N$ is the number of neutrons. Here, we have assumed that $\sin^2\hat{\theta}_W$ evaluated at the relevant APV momentum transfer of $Q \approx 2.4$ MeV is, to good accuracy, the same as $\hat{s}_0^2$; see Ref. [83] for further details. Radiative corrections to Equation (21) come from the $W$ and $Z$ boson self-energies, the $\gamma Z$ mixing renormalization of $\sin^2\theta_W$, and the so-called $WW$, $ZZ$, and $\gamma Z$ box diagrams [22–25] also depend on $M_W$, either directly $\sim \ln M_W^2$ or indirectly via the value of the strong coupling constant evaluated at $M_W$.

The interaction (20) mixes atomic states with opposite parities and thus gives rise to the otherwise forbidden electric–dipole transitions between two states with the same nominal parity, e.g., $6S_{1/2} \to 7S_{1/2}$ in $^{133}$Cs. A measurement of the amplitude of such a transition leads to the extraction of the values $Q_W$ and $\hat{s}_0^2$, which may then be compared with the SM predictions to give hints about new physics. In the next section, we present several mechanisms through which new physics effects may alter the SM value for the nuclear weak charge $Q_W$.

### 2.2. New Physics Contributions to Atomic Parity Violation

In this section, we consider three beyond-SM contributions to the nuclear weak charge $Q_W$, namely, a tree-level exchange of a new $Z'$ boson, corrections to the $Z$ and $W$ boson self-energies through quantum loops involving beyond-SM particles, and SM particles coupling to a new dark $Z_d$ boson. Although the results presented here are not new, they serve as a basis for our discussions in Section 4.

Let us begin with the tree-level correction due to the exchange of a new heavy $Z'$ boson, which appears in several extensions of the SM, including $SO(10)$, $E_6$, and extradimensional theories [84–91]. In the low energy limit, this yields an additional term similar to the effective Lagrangian (12).

$$\mathcal{L}_{\mathrm{NC}}^{\prime eq} = \frac{1}{M_{Z'}^2}\bar{e}\gamma_\mu\gamma_5 e\left(f_{AV}^{eu}\bar{u}\gamma^\mu u + f_{AV}^{ed}\bar{d}\gamma^\mu d\right),\tag{22}$$

where $f_{AV}^{eu}$ and $f_{AV}^{ed}$ are the axial–electron–vector–quark coupling constants for the exchange of the new $Z'$ boson of mass $M_{Z'}$ [92,93]. By integrating out the quark degrees of freedom, we obtain the following contribution to $Q_W$

$$\Delta Q_W^{Z'} = -\frac{2\sqrt{2}}{G_F}\frac{\bar{f}_{AV}^{eq}}{M_{Z'}^2}3(Z+N),\tag{23}$$

where the average electron-quark coupling $\bar{f}_{AV}^{eq}$ is defined as

$$\bar{f}_{AV}^{eq} = \frac{f_{AV}^{eu}(2Z+N) + f_{AV}^{ed}(Z+2N)}{3(Z+N)}.\tag{24}$$

In a simple model where $Z'$ has the same couplings as $Z$, Equation (23) reduces to [26,27,94]

$$\Delta Q_W^{Z'} = -Q_W^{SM} \frac{M_Z^2}{M_{Z'}^2} \,. \tag{25}$$

We note that the analysis above applies only to the case where the mass of the new $Z'$ boson is much larger than the relevant momentum transfer ($Q \approx 2.4$ MeV in the case of APV in $^{133}$Cs). If, on the other hand, $M_{Z'} \ll Q$, then the local approximation for the $Z'$ boson propagator leading to the contact interaction (22) is no longer valid. Instead, the exchange of a light $Z'$ boson gives rise to a (long range) Yukawa-like PV interaction [92,93].

$$\Delta H_W' = -\frac{G_F}{2\sqrt{2}} \bar{e}(\mathbf{r})\gamma_5 e(\mathbf{r}) \Delta Q_W \frac{M_{Z'}^2}{4\pi r} e^{-M_{Z'} r} \rho(\mathbf{r}) \,. \tag{26}$$

The effect of this on $Q_W$ is equivalent to multiplying Equation (23) by a factor

$$K(M_{Z'}) = \frac{\int \langle \bar{e}(\mathbf{r})\gamma_5 e(\mathbf{r})\rangle \frac{M_{Z'}^2 e^{-M_{Z'} r'}}{4\pi r'} \rho(\mathbf{r}')d^3r d^3r'}{\langle \bar{e}(\mathbf{r})\gamma_5 e(\mathbf{r})\rangle \rho(\mathbf{r}')d^3r d^3r'} \,, \tag{27}$$

which accounts for the long range nature of the interaction. For $M_{Z'} \approx 2.4$ MeV, $K(M_{Z'}) = 1/2$.

Next, we consider vacuum polarization effects of the self-energies $\Pi_{WW}(q^2)$ and $\Pi_{ZZ}(q^2)$ of the $W$ and $Z$ bosons. These effects may arise, for example, from quantum loops involving supersymmetric [95] or technicolor particles [96]. The form of Equation (17) is especially convenient for including these contributions. Regardless of their nature, if the new physics phenomena are associated with very large energy scales, their effects at low energies may be described solely by the weak isospin-conserving parameters $S_W$ and $S_Z$ and the weak isospin-breaking parameters $T$ defined by [26,28,97].

$$\frac{\Pi_{WW}^{\mathrm{new}}(0)}{M_W^2} - \frac{\Pi_{ZZ}^{\mathrm{new}}(0)}{M_Z^2} = \alpha(M_Z)T \,, \tag{28}$$

$$\left[\frac{\Pi_{WW}^{\mathrm{new}}(M_W^2) - \Pi_{WW}^{\mathrm{new}}(0)}{M_W^2}\right]_{\overline{\mathrm{MS}}} = \frac{\alpha(M_Z)}{4\hat{s}_0^2} S_W \,, \tag{29}$$

$$\left[\frac{\Pi_{ZZ}^{\mathrm{new}}(M_Z^2) - \Pi_{ZZ}^{\mathrm{new}}(0)}{M_Z^2}\right]_{\overline{\mathrm{MS}}} = \frac{\alpha(M_Z)}{4\hat{s}_0^2(1-\hat{s}_0^2)} S_Z \,, \tag{30}$$

where $\Pi_{WW}^{\mathrm{new}}$ and $\Pi_{ZZ}^{\mathrm{new}}$ are new physics vacuum polarization contributions, and $\alpha(M_Z) \approx 1/127.94$ is the fine-structure constant at $M_Z$.

To the leading order, the new physics effects modify the $^{133}$Cs weak charge by contributing to the radiative corrections to $G_F$ and $\hat{s}_0^2$. The contribution to $G_F$ may be conveniently expressed via a multiplicative factor, $G_F \to \rho^{\mathrm{new}} G_F$, where

$$\rho^{\mathrm{new}} = 1 + \frac{\Pi_{WW}^{\mathrm{new}}(0)}{M_W^2} - \frac{\Pi_{ZZ}^{\mathrm{new}}(0)}{M_Z^2} = 1 + \alpha(M_Z)T \approx 1 + 0.00782T \,, \tag{31}$$

by virtue of Equation (28). Assuming that $S_W = S_Z = S$, the quantity $\Delta \hat{r}_W$ in Equation (17) receives an additional contribution [26].

$$\Delta \hat{r}_W^{\mathrm{new}} = \left[\frac{\Pi_{ZZ}^{\mathrm{new}}(M_Z^2) - \Pi_{WW}^{\mathrm{new}}(0)}{M_W^2}\right]_{\overline{\mathrm{MS}}} = \frac{\alpha(M_Z)}{4\hat{s}_0^2(1-\hat{s}_0^2)} S - \alpha(M_Z)T \,, \tag{32}$$

which, when solved for $\hat{s}_0^2$ perturbatively, gives

$$(\Delta \hat{s}_0^{\mathrm{new}})^2 = 0.00365S - 0.00261T \,. \tag{33}$$

By using Equations (31) and (33) in Equation (21), one finds the beyond-SM contribution the $^{133}$Cs weak charge as [98].

$$\Delta Q_W^{ST}\left(^{133}\text{Cs}\right) = -0.800S - 0.007T \,. \tag{34}$$

where the suppression of the $T$-contribution is a result of an accidental cancellation between $\rho^{\text{new}}$ and $\Delta \hat{r}_W^{\text{new}}$ particular to $^{133}$Cs. Equation (34) shows that the $^{133}$Cs APV experiment is sensitive to $S$ but not $T$.

Finally, we consider the effects of a dark $Z_d$ boson of mass $M_{Z_d} \sim$ MeV-GeV, which couples to the SM via kinetic mixing with the photon and mass mixing with the $Z$ boson. Such a particle arises from an extra $U(1)_d$ gauge symmetry and is a prominent dark matter candidate (the dark photon) [99–102]. The extended QED Lagrangian with the new $U(1)_d$ included reads [103].

$$\mathcal{L}_{Z_d} = -\frac{Z_d^{\mu\nu}Z_{d\mu\nu}}{4} - M_{Z_d}^2 Z_d^\mu Z_{d\mu} + \frac{\epsilon B_{\mu\nu}Z_d^{\mu\nu}}{2\cos\theta_W} \,, \tag{35}$$

where $B_{\mu\nu} = \partial_\mu B_\nu - \partial_\nu B_\mu$ and $Z_{d\mu\nu} = \partial_\mu Z_{d\nu} - \partial_\nu Z_{d\mu}$ are the field strengths for the $U(1)_Y$ and $U(1)_d$ vector fields, respectively. The kinetic mixing term in Equation (35) may be removed by shifting $B_\mu \to B_\mu + \frac{\epsilon}{\cos\theta_W}Z_{d\mu}$, with which the photon and $Z$ boson fields become $A_\mu \to A_\mu + \epsilon Z_{d\mu}$ and $Z_\mu \to Z_\mu - \epsilon \tan\theta_W Z_{d\mu}$. As a result of this field redefinition, an induced coupling of $Z_d$ to the SM electromagnetic current $J_{\text{em}}^\mu$ appears and has the form

$$\mathcal{L}_{\text{em}}^{Z_d} = -e\epsilon J_{\text{em}}^\mu Z_{d\mu} \,. \tag{36}$$

Neutral-current couplings of $Z_d$ to the SM sector may be included by introducing a $Z - Z_d$ mass mixing

$$\mathcal{L}_{\text{mass mixing}} = -\epsilon_Z M_Z^2 Z^\mu Z_{d\mu} \,, \tag{37}$$

with a mixing coefficient $\epsilon_Z = (M_{Z_d}/M_Z)\delta$, where $\delta$ is a small model-dependent quantity [30,104]. The vacuum oscillations between the $Z$ and $Z_d$ fields due to mass mixing may be removed by another field redefinition, after which $Z_d$ couples directly to the SM sector via

$$\mathcal{L}_{\text{NC}}^{Z_d} = -\frac{g\epsilon_Z}{\cos\theta_W} J_Z^\mu Z_{d\mu} \,, \tag{38}$$

where $J_Z^\mu$ is again the weak neutral current appearing in Equation (9).

Both interactions (36) and (38) contribute to parity violation due to electron-quark coupling by exchanging a $Z_d$ boson. As with the exchange of a $Z$ boson, the electron vertex is the axial part of Equation (38). The quark vertex, on the other hand, can either be the vector part of Equation (38) (as with a $Z$ boson) or the electromagnetic coupling (36). The overall effect of the interactions (36) and (38) for APV thus appears as a modification of the nuclear weak charge (assuming that $M_{Z_d}$ is much larger than the momentum exchange) [30,104]

$$\Delta Q_W^{Z_d} = \delta^2 Q_W^{SM} + 4Z\frac{\epsilon}{\epsilon_Z}\delta^2 \cos\hat{\theta}_W \sin\hat{\theta}_W \,. \tag{39}$$

The quantities $\bar{f}_{AV}^{eq}/M_{Z'}^2$, $S$, $\delta$ and $\epsilon/\epsilon_Z$ parameterizing the new physics contributions to APV have been constrained by comparing the measured values of $Q_W(^{133}\text{Cs})$ with the SM prediction. In the next section, we consider the recent result from the CDF II Collaboration for $M_W$ which shows a significant $7\sigma$ tension with the SM [33]. We shall assume that all other quantities except $\sin^2\hat{\theta}_W$ remain the same. By the virtue of Equation (17), the shift in $M_W$ then implies a corresponding change in the value of $\sin^2\hat{\theta}_W(M_Z)$ and thus of $Q_W(^{133}\text{Cs})$ via $\hat{s}_0^2$. We will use this "new" SM value for $Q_W(^{133}\text{Cs})$ to adjust existing constraints on $\bar{f}_{AV}^{eq}/M_{Z'}^2$, $S$, $\delta$ and $\epsilon/\epsilon_Z$.

## 3. Numerical Results

The value of $\hat{s}_0^2$ as predicted by the SM may be related to the $W$ boson mass $M_W$ via Equation (17) and the running of the weak angle [24,105,106].

$$\sin^2\hat{\theta}_W(Q^2) = \kappa(Q^2)\sin^2\hat{\theta}_W(M_Z) = \frac{\kappa(Q^2)A^2}{M_W^2(1-\Delta\hat{r}_W)}. \tag{40}$$

In this paper, we assume the standard value of $\kappa(0) \approx 1.03$ [22,23]. Equation (40) then shows that $\hat{s}_0^2$ is inversely proportional to $M_W^2$ (strictly speaking $\Delta\hat{r}_W$ also depends on $M_W$ via $\sin^2\hat{\theta}_W(M_Z)$; however, since $\Delta\hat{r}_W \approx 0.06994 \ll 1$, we can safely ignore this dependence).

Equations (21) and (40) show the dependence of the SM value for $Q_W$ on the physical mass $M_W$. Since the value of $M_W$ is such that $\hat{s}_0^2 \approx 1/4$, the dependence is relatively weak for heavy atoms where $N$ is large. Nevertheless, the extraordinary accuracy of APV experiments means that the measured weak charge could be sensitive to variations in the experimental value of $M_W$. It is also worth noting that the suppression due to the neutron is absent in the case of a proton, whose weak charge $Q_W(p) = -0.0719(45)$ [75] has an enhanced sensitivity to $\hat{s}_0^2$. Thereby, renewed efforts on the atomic hydrogen APV experiment would be of great interest as an independent indirect probe of $M_W$ mass.

Let us now consider the most recent result from the CDF II experiment at Tevatron [33], which found $M_W = 80{,}433.5 \pm 9.4\,\text{MeV}/c^2$, equivalent to a $7\sigma$ deviation from the SM model value of $M_W = 80{,}357 \pm 6\,\text{MeV}/c^2$. Ref. [33] also obtained a value of $M_Z = 91{,}192 \pm 7.5\,\text{MeV}/c^2$, which is consistent with the world average of $M_Z = 91{,}187 \pm 2.1\,\text{MeV}/c^2$. Plugging these values into Equations (21) and (40) and the formulae for radiative corrections [46], while assuming that all other parameters are unchanged, we find that the stand-alone CDF II result for $M_W$ implies

$$Q_W^{\text{CDF II}}\left(^{133}\text{Cs}\right) = -73.11(1). \tag{41}$$

Our revised $^{133}$Cs weak charge (41) corresponds to a shift of 0.16% or $8.5\sigma$ from the "old" SM value of $Q_W^{\text{SM}}(^{133}\text{Cs}) = -73.23(1)$. Our approach should be contrasted with global fits with $M_W = 80{,}433.5 \pm 9.4\,\text{MeV}/c^2$, which do not show an appreciable change in the value of $\sin^2\hat{\theta}_W$, but rather slight variations in a wide array of EW parameters [34–37]. We believe that our approach makes the role of $M_W$ more explicit and eliminates a potential bias.

Compared to the value extracted from Ref. [16], $Q_W^{2010}(^{133}\text{Cs}) = -73.16(29)_{\text{exp}}(20)_{\text{th}}$ and the result $Q_W^{2012}(^{133}\text{Cs}) = -72.58(29)_{\text{exp}}(32)_{\text{th}}$ extracted from Ref. [17] (the superscripts 2010 and 2012 denote the year of the corresponding publications), we have

$$Q_W^{2010} - Q_W^{\text{CDF II}} = -0.05(35), \tag{42}$$

$$Q_W^{2012} - Q_W^{\text{CDF II}} = 0.53(43), \tag{43}$$

where the uncertainties in Equations (42) and (43) were obtained by adding the corresponding theoretical and experimental errors and the uncertainty in Equation (41) in quadrature. From this, one observes that the result of Ref. [16] is $0.14\sigma$ smaller than the CDF II value while the result of Ref. [17] is $1.2\sigma$ larger. The comparison between these values is presented in Figure 1. Clearly, both results agree well with our revised SM value (41) within their respective error bars, while their average is in excellent agreement with (41).

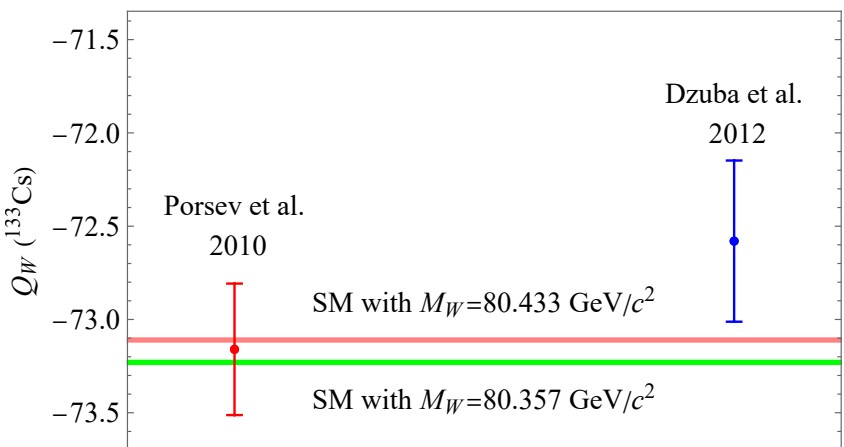

**Figure 1.** (Color online) Comparison between the $^{133}$Cs nuclear weak charge as predicted by the Standard Model (SM) with the mass of the $W$ boson being $M_W = 80.357\,\text{GeV}/c^2$ [82] (blue band), the SM with $M_W = 80.433\,\text{GeV}/c^2$ [33] (pink band), and $^{133}$Cs APV experiment [1] with different calculations for the $^{133}$Cs atomic structure factor (red and blue points).

Another substantial effect of the new value of $M_W$ is on APV in atomic hydrogen due to the absence of the otherwise leading and $M_W$-independent neutron contribution to nuclear weak charge. Indeed, $M_W = 80{,}433.5 \pm 9.4\,\text{MeV}/c^2$ [33] implies a revised SM value for the proton weak charge

$$Q_W^{\text{CDFII}}(p) \approx 0.0730(21)\,, \tag{44}$$

corresponding to a shift of 2.7% or $0.9\sigma$ away from the current SM prediction of $Q_W^{\text{SM}}(p) = 0.0711(2)$. It is also 1.5% or $0.22\sigma$ away from the nuclear physics measurement $Q_W(p) = 0.0719(45)$ [75]. A 1% measurement of a proton weak charge in APV experiments with hydrogen [107,108] may provide further evidence for or against the new CDF II result.

Using the new value for $Q_W^{\text{SM}}$ (41), we can derive new constraints for the parameters of a new $Z'$ boson appearing in Equation (23) as

$$\left(\frac{\bar{f}_{AV}^{eq}}{M_{Z'}^2}\right)^{2010} = (5.2 \pm 3.6) \times 10^{-9}\,\text{GeV}^{-2}\,, \tag{45}$$

$$\left(\frac{\bar{f}_{AV}^{eq}}{M_{Z'}^2}\right)^{2012} = (-5.5 \pm 4.5) \times 10^{-9}\,\text{GeV}^{-2}\,, \tag{46}$$

where, again, the superscript 2010 corresponds to Ref. [16], and the superscript 2012 corresponds to Ref. [17]. If we assume that the new $Z'$ has the same couplings as $Z$, then by using Equation (25) and the value $Q_W^{2012}$, which results in a positive pull away from the SM, we obtain a direct constraint on $M_{Z'}$ as

$$M_{Z'} \geq 1.1\,\text{TeV}/c^2\,, \tag{47}$$

which is comparable to limits set by other electroweak precision data [109,110], interference effects at LEP-II [111], and the Tevatron [112].

Similarly, by using Equation (34), we can obtain constraints on the oblique parameter $S$ of new vacuum polarization effects as

$$S^{2010} = 0.06(44)\,, \tag{48}$$

$$S^{2012} = -0.66(54)\,. \tag{49}$$

Finally, by using Equation (39), we derive updated constraints on the kinetic and mass mixing parameters of a dark $Z_d$ boson as

$$\left[ \delta^2 \left( 1 - 1.28 \frac{\epsilon}{\epsilon_Z} \right) \right]^{2010} = 0.00684(3) \,, \tag{50}$$

$$\left[ \delta^2 \left( 1 - 1.28 \frac{\epsilon}{\epsilon_Z} \right) \right]^{2012} = -0.00725(4) \,. \tag{51}$$

## 4. Discussions

We have demonstrated how the recent measurement of the $W$ boson mass [33] meaningfully affects the interpretation of APV experiments. More specifically, the new $M_W$ boson mass implies a 0.16% shift in the SM prediction for the $^{133}$Cs weak charge and a 2.7% shift in the prediction for the proton weak charge. We find that the new value for the $^{133}$Cs nuclear weak charge reconciles the tension between the two most recent results for $Q_W(^{133}\text{Cs})$ extracted from the same experiment [1] but with different methods [16,17] of computing the atomic structure factor $k_{\text{PV}}$.

This reconciliation does not, however, signify an end to the story of APV. In fact, the disagreement between the two results [16,17], in particular the opposite signs of their estimates for the so-called core contribution, remains unexplained. Furthermore, since the uncertainties in these two results overlap one another and the new SM prediction (41) (see Figure 1), they provide little evidence for or against the new measurement of the new $M_W$ boson mass. Such a confirmation or refutation may be possible, however, if the error bars in Figure 1 are reduced by half, i.e., to the level of $\lesssim 0.2\%$. In this sense, our result for $^{133}$Cs provides further motivation for reducing the uncertainty in atomic calculations for APV [21]. It is worth mentioning that new measurements of electric dipole transition amplitudes in $^{133}$Cs have recently been carried out at the level of 0.1~0.2% uncertainty [113–115]. These results serve as useful standards for gauging the accuracy of theoretical atomic calculations. Similarly, our results for the proton weak charge motivate next-generation APV experiments in hydrogen.

An alternative APV approach is the measurements of APV in a chain of isotopes, which forgoes the evaluation of atomic structure factors $k_{\text{PV}}$ altogether. Such measurements yield ratios of weak charges, $Q_W / Q'_W$, of two nuclei with a fixed number of protons $Z$ but differing number of neutrons ($N$ and $N' = N + \Delta N$), see, e.g., Ref. [62]. In the isotopic-chain measurements, the sensitivity to $M_W$ (through $\hat{s}_0^2$) can be expressed as

$$\frac{Q_W / N}{Q'_W / N'} \approx 1 - \frac{\Delta N}{N} \frac{Z}{N} \left( 1 - 4\hat{s}_0^2 \right) \,, \tag{52}$$

while in a single-isotope measurement, the relevant quantity is

$$\frac{Q_W}{-N} = 1 - \frac{Z}{N} \left( 1 - 4\hat{s}_0^2 \right) \,. \tag{53}$$

Comparing the two expressions, we see that in the isotopic-chain measurements, there is an extra suppression factor of $\Delta N / N$, which is $\ll 1$ for heavy nuclei of practical interest. Thereby, single-isotope measurements are more sensitive to varying $M_W$ than the isotopic-chain experiments.

Throughout this paper, we have focused on how the new $M_W$ measurement affects the SM prediction for the nuclear weak charges and thus limits new physics by changing the value of $\hat{s}_0^2$. Evidence for beyond-SM contributions may also be constrained by direct low-energy precision measurements of the weak mixing angles themselves. Indeed, significant improvement in the short term is likely to come from PV polarized-electron scattering asymmetries at MESA [116] and Jefferson Lab [117] rather than from APV. Nevertheless, as noted above, new APV experiments and theoretical calculations at the level of 0.2% uncertainty will have meaningful long-term contributions to testing the SM at low energies. In

Figure 2, we plot the running of $\sin\hat{\theta}_W$ with the $W$ boson mass $M_W = 80.357\,\text{GeV}/c^2$ [82] and $M_W = 80.433\,\text{GeV}/c^2$ [33]. The change in $M_W$ induces a downward $7.5\sigma$ shift in $\sin\hat{\theta}_W$ across the energy scale up to 1 TeV. It may be observed that the blue curve, corresponding to $M_W = 80.357\,\text{GeV}/c^2$, gives a better fit to experimental measurements at the $Z$-pole, while the red curve, corresponding to $M_W = 80.433\,\text{GeV}/c^2$, gives a somewhat better fit to low-energy measurements. We reemphasize that this result is obtained by considering the new CDF-II value for $M_W = 80{,}433.5 \pm 9.4\,\text{MeV}/c^2$ while keeping all other SM parameters fixed. Global fits that include the new value for $M_W$ do not show an appreciable change in the value of $\sin^2\hat{\theta}_W$, but rather slight variations in a wide array of EW parameters [34–37]. We believe that our approach makes the role of $M_W$ more explicit and eliminates a potential bias.

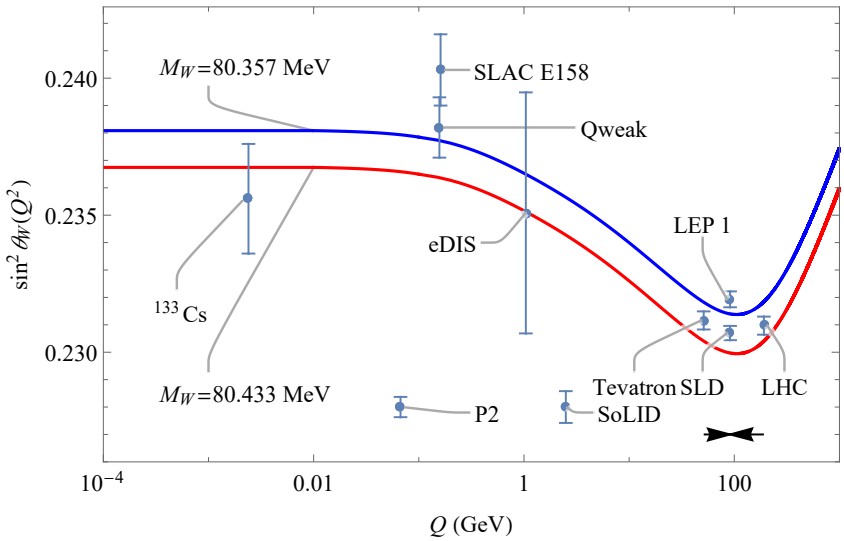

**Figure 2.** (Color online) Running of $\sin\hat{\theta}_W$ predicted by the Standard Model (SM) with the mass of the $W$ boson being $M_W = 80.357\,\text{GeV}/c^2$ [82] (blue line) and the SM with the new value $M_W = 80.433\,\text{GeV}/c^2$ [33] (red line). Results of several low-energy parity-violating lepton scattering experiments, as well as $Z$-pole measurements are also presented. The points P2 and SoLID are projected values from MESA's P2 experiment [116] and Jefferson Lab's SoLID experiment [117]. For clarity, the Tevatron and LHC results have been horizontally shifted.

## 5. Conclusions

In this paper, we showed that the stand-alone CDF-II value for the mass of the W-boson implies a 0.16% or $8.5\sigma$ shift in the Standard Model (SM) prediction of the $^{133}$Cs weak charge. This shift, if exists, is potentially detectable by atomic parity violation (APV) experiments on $^{133}$Cs with uncertainties $\lesssim 0.1{\sim}0.2\%$. Effort to reduce the theoretical uncertainties of atomic calculation down to this level is in progress [21], whereas new APV experiments on $^{133}$Cs are being planned [4,5]. Using the new SM prediction for the $^{133}$Cs weak charge, we readjusted APV constraints on parameters of physics beyond the SM, such as the mass of a new $Z'$-boson, Equation (47), the oblique parameter of new vacuum polarization effects, Equation (48), and the kinetic and mass mixing parameters of a dark $Z_d$ boson, Equation (50). We also showed that the stand-alone CDF-II value for the mass of the W-boson implies a 2.7% shift in the prediction for the proton weak charge, thus motivating new APV experiments in hydrogen.

**Author Contributions:** Formal analysis, H.B.T.T. and A.D.; Investigation, H.B.T.T.; Writing—original draft, H.B.T.T. and A.D.; Writing—review & editing, H.B.T.T. and A.D.; Supervision, A.D.; Project administration, A.D.; Funding acquisition, A.D. All authors have read and agreed to the published version of the manuscript.

**Funding:** This research was funded by the U.S. National Science Foundation grants PHY-1912465 and PHY-2207546, by the Sara Louise Hartman endowed professorship in Physics, and by the Center for Fundamental Physics at Northwestern University.

**Data Availability Statement:** Not applicable.

**Acknowledgments:** The authors thank Igor Samsonov and Xiaochao Zheng for helpful discussions. We thank Jens Erler for providing his code for computing radiative corrections.

**Conflicts of Interest:** The authors declare no conflict of interest.

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
