# Peer review of "Implications of W-Boson Mass Anomaly for Atomic Parity Violation"

_atoms, doi:10.3390/atoms10040149_

Round 1

Reviewer 1 Report

Comments;

The WM Anomaly recently detected by CDF II is a hot topic and has required a thorough analysis from all sorts of aspects. In this respect, the actual paper attempted to relate the CDF II anomaly to physics beyond the Standard Model (SM). It seems logical that the atmospheric parity violation might affect the change in the SM. The idea seems good and the paper should be published in the journal. I have some comments to add by the authors to improve the quality.

1.  Why APV experiment is  NOT sensitive to T parameter?

2. Are there any constraints from LEP on the mixing angle between Z and Dark Z?

Author Response

Dear Referee,

Thank you for reviewing our manuscript and providing useful feedbacks.

In accordance with your suggestions:

  1. We have included a detailed discussion of the new Physics contribution to the oblique parameters S and T. This also allows us to explain why 133Cs APV is not sensitive to T (due to an accidental cancellation). Please see the part containing Eqs. (23) - (28) in the updated manuscript.
  2. The constraint we obtained for the Z' boson is comparable to that imposed by LEP-II, i.e., mass of Z'~ 1 TeV. We have added this comment to after Eq. (39) (see updated manuscript).

(*) Please note that we have updated the numbers in our paper (see Eqs. (37) to (41)). This is a result of including radiative corrections to the calculations. These changes do not affect the main conclusion of the paper.

Reviewer 2 Report

In this article, the authors discuss the implications of the new W mass measurement to the atomic parity violation. This is a sound and very timely analysis. I have one suggestion for authors. The authors discuss one example of the new physics contribution from the Z' particle. The analysis will be more complete if they comment on related Z' phenomenology. 

Author Response

Dear Referee,

Thank you very much for reviewing our manuscript.

From the Referee's suggestion, we have included a new subsection (see Sect. 2.2: New physics contributions to atomic parity violation) where we discuss in details Z' phenomenology. In this section, we also discuss other new physics contributions to APV such as new vacuum polarization effects and new dark boson.

We believe this satisfies the Referee's recommendation and makes our paper more informative for the readers.

Reviewer 3 Report

The article is well written and the logic clear. In addition to the detailed Discussion section, a new section with Conclusions is recommended.

Given the current significances, including a further discussion on the potential for an increase of sensitivity could be beneficial for the reader.

I recommend publication with minor amendments.

Author Response

Dear Referee,

Thank you for reviewing our manuscript and providing useful suggestions.

From the Referee's suggestions, we have included a "Conclusions" section to summarize our findings.

We have also added more comments and references on the current progress and future potential to improve the sensitivity of 133Cs APV (see line 170 in the new draft).
